# Global disruption of coral broadcast spawning associated with artificial light at night

Thomas W. Davies [1] ✉, Oren Levy [2,3], Svenja Tidau[1],
Laura Fernandes de Barros Marangoni [4], Joerg Wiedenmann[5],
Cecilia D'Angelo[5] & Tim Smyth [6]

Coral broadcast spawning events - in which gametes are released on certain nights predictably in relation to lunar cycles - are critical to the maintenance and recovery of coral reefs following mass mortality. Artificial light at night (ALAN) from coastal and offshore developments threatens coral reef health by masking natural light:dark cycles that synchronize broadcast spawning. Using a recently published atlas of underwater light pollution, we analyze a global dataset of 2135 spawning observations from the 21st century. For the majority of genera, corals exposed to light pollution are spawning between one and three days closer to the full moon compared to those on unlit reefs. ALAN possibly advances the trigger for spawning by creating a perceived period of minimum illuminance between sunset and moonrise on nights following the full moon. Advancing the timing of mass spawning could decrease the probability of gamete fertilization and survival, with clear implications for ecological processes involved in the resilience of reef systems.

Coral reefs are among the most biodiverse[1], economically important[2], and threatened[3] ecosystems. Climate change induced mass bleaching events[4,5], habitat destruction, fisheries, and pollution combined have reduced coral reef cover substantially since the 1950s[6]. The complete loss of tropical corals is anticipated over the next 100 years[7]. Addressing the causes of coral reef loss is challenging as ocean temperatures continue to rise even under optimistic Intergovernmental Panel on Climate Change (IPCC) scenarios[8]. Sustaining future coral reefs may well require the maintenance of healthy coral reproduction for their recovery between mass mortality events and recruitment into newly suitable habitats. Synchronized broadcast spawning is a reproductive strategy that involves the release of a large number of eggs and sperm simultaneously by a large proportion of a local coral population on specific nights of the year. Spawning synchronization maximizes reproductive contact between conspecifics[9] and is widely considered

to ensure optimal environmental conditions for the dispersal, development, and recruitment of coral larvae. Disruptions in the timing of broadcast spawning have recently been reported from coral reefs in the northern Red Sea[10,11]. The global extent of this disruption and its causes remain unclear.

Moonlight cycles entrain spawning synchronization[12–14]. ALAN from coastal developments disrupts these cycles presenting a potential threat to coral reproduction[15]. ALAN has demonstrated impacts on coral gamete development and spawning in experimental settings[14,16]. Whether ALAN disrupts coral spawning in the real world is currently unquantified.

Here we show that disruption of broadcast spawning by corals around the world is associated with exposure to artificial light at night. We analyze the recently published "Global atlas of artificial light at night under the sea"[17] and the Coral Spawning Database[18,19] to establish

[1]School of Biological and Marine Sciences, University of Plymouth, Drake Circus, Plymouth, Devon PL4 8AA, UK. [2]Mina and Everard Goodman Faculty of Life Sciences, Bar-Ilan University, Ramat Gan 52900, Israel. [3]Israel The H. Steinitz Marine Biology Laboratory, The Interuniversity Institute for Marine Sciences of Eilat, P.O. Box 469 Eilat 88103, Israel. [4]Smithsonian Tropical Research Institute, Smithsonian Institution, Ciudad de Panama 0843-03092, Panama. [5]Coral Reef Laboratory, University of Southampton, European Way, Southampton SO143ZH, UK. [6]Plymouth Marine Laboratory, Prospect Place, Plymouth, Devon PL1 3DH, UK. ✉e-mail: thomas.w.davies@plymouth.ac.uk

whether broadcast spawning by corals is disrupted in ALAN-impacted waters (Fig. 1). The 1 km resolution global atlas of artificial light under the sea represents the critical depth to which ALAN can cause biological impacts (Methods). We quantify the impact of underwater light pollution on the timing of broadcast spawning in Days of Spawning Relative to the Nearest Full moon (DoSRtNF) using 2135 coral spawning observations recorded from 156 species representing 12 genera across 52 locations in 19 ecoregions around the world between the years 2000 and 2019 (Fig. 1). In our analysis, each observation in the coral spawning database is classified as exposed to underwater ALAN ("Lit") where the critical depth of ALAN exceeded the depth of the observation or−where no depth was recorded−the minimum depth at which the species is found (from the Coral Trait Database[20]). Where this was not the case, each observation is classified as not exposed to ALAN ('Unlit'). Our results indicate that corals exposed to ALAN are spawning between one and three days closer to the full moon compared to those not exposed to ALAN. Shifting the timing of broadcast spawning may reduce the survival and fertilization success of gametes, and genetic connectivity between neighboring lit and unlit reef systems.

## Results

We tested whether underwater ALAN disrupted the timing of broadcast spawning relative to the full moon. Our analysis accounted for taxonomic differences (Genus); the rate changes in average annual nighttime sea surface temperature ($\Delta$SST) and water clarity (diffuse attenuation coefficient at 490 nm, $\Delta Kd_{490}$) over the year 2003 to 2022 period; latitudinal effects (Distance from the Equator, DfE); differences in spawning times between ecoregions (Ecoregion[19]); and spatial autocorrelation between sampling locations. Model selection was performed on all nested versions of a global spatially autocorrelated generalized linear mixed effects model [DoSRtNF-ALAN*Genus + ALAN*$\Delta$SST + ALAN*$\Delta Kd_{490}$ + ALAN*DfE + Ecoregion + Matern(1|Longitude + Latitude)] fitted with a Poisson error distribution (dispersion = 0.82). ALAN impacts on DoSRtNF differed between genera in all 20 top-ranked models attaining a cumulative probability of 0.81 (Supplementary Table 1). The ALAN*$\Delta Kd_{490}$ interaction was not included as an explanatory term in the top four ranked models that were within two $\Delta$AIC points of the top-ranked model. The remaining terms were included in at least one of the four top-ranked models and were selected for further interrogation using Likelihood Ratio Tests [DoSRtNF-ALAN*Genus + ALAN*$\Delta$SST + ALAN*DfE + Ecoregion + $\Delta Kd_{490}$ + Matern(1|Longitude + Latitude)]. The resulting selected model explained significantly more variance in DoSRtNF compared to a null intercept-only model ($\chi^2 = 171.52$, $p < 0.001$).

ALAN impacts on the timing of broadcast spawning differed with each genus (ALAN:Genus $\chi^2 = 44.00$, $p < 0.001$, Fig. 2a), but not with differences in the rate of change in sea surface temperature (ALAN:$\Delta$SST $\chi^2 = 0.11$, $p = 0.746$), and distance from the equator (ALAN: DfE $\chi^2 = 3.57$, $p = 0.059$) between spawning locations. Differences in the rate of change in water clarity and sea surface temperature between spawning locations also had no direct impacts on the timing of coral broadcast spawning ($\Delta Kd_{490}$ $\chi^2 = 2.46$, $p = 0.117$; $\Delta$SST $\chi^2 = 0.75$, $p = 0.387$). The timing of broadcast spawning in relation to the full moon was, however, dependent on the Ecoregion in which spawning was observed (Ecoregion $\chi^2 = 37.25$, $p < 0.01$, Fig. 2c), with corals located closer to the equator spawning closer to the full moon compared to those located at higher latitudes (DfE $\chi^2 = 5.25$, $p < 0.05$, Fig. 2b).

Broadcast spawning occurred closer to the full moon on lit versus unlit reefs in ten out of the 12 genera analysed (Fig. 2a and Supplementary Table 2). Corals of the genera *Montipora* spp. and *Favites* spp. spawned on average one day closer to the full moon on lit compared to unlit reefs (Fig. 2a and Supplementary Table 3). *Acropora* spp., *Dipsastraea* spp., *Goniastrea* spp., *Galaxea* spp., *Acanthastrea* spp., *Platygyra* spp., and *Cyphastrea* spp. spawned on average 2 days closer, and *Porites* spp. 3 days closer to the full moon on lit compared to unlit reefs. These results were significant at the 95% confidence level or greater (Supplementary Table 2). *Echinophyllia* spp. spawned one day closer to the full moon on lit compared to unlit reefs (Supplementary Table 3), a difference that was significant at the 90% confidence level (Fig. 2a and Supplementary Table 2). Our analysis indicates these impacts are likely to be globally widespread on coral reefs exposed to ALAN (Fig. 1).

The first extended period of stable light intensity between sunset and moonrise following the full moon has recently been implicated as a natural synchronizer for broadcast spawning several days later[12]. The onset of this period occurs in the days following a full moon, when it rises progressively later after sunset. The presence of ALAN may advance the day on which this period of minimum light intensity occurs by artificially brightening the night sky such that declines in solar irradiance during the latter stages of twilight are no longer detectable. We modeled the sea surface light regimes of a high

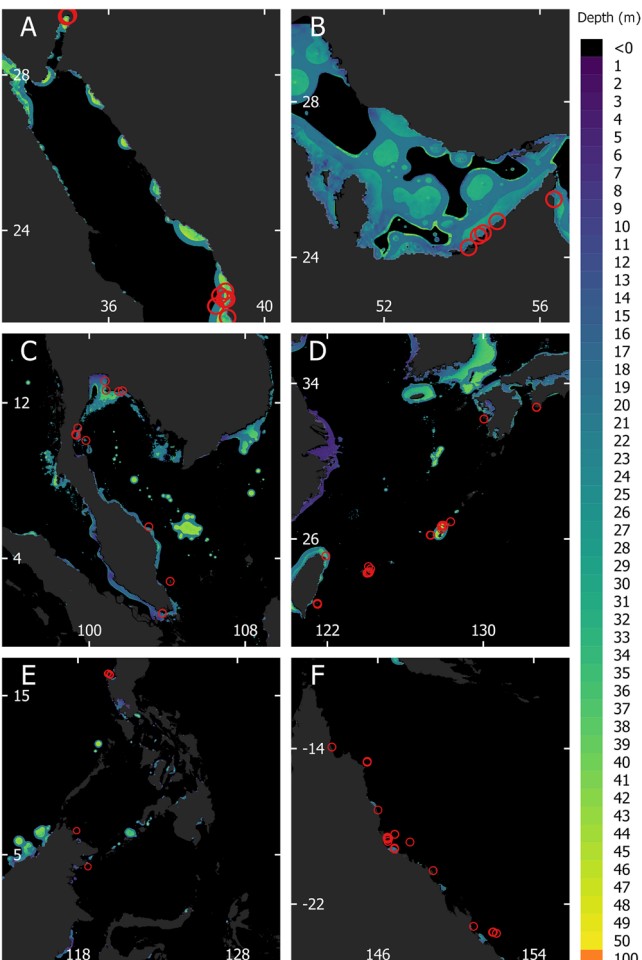

**Fig. 1 | 21st-century coral broadcast spawning observation locations in relation to underwater light pollution from coastal developments.** Light exposure is defined as the average depth of biologically important ALAN throughout the year 2020 (Methods). **A** The Red Sea and the Gulf of Eliat; **B** The Persian Gulf; **C** The Gulf of Thailand and the South China Sea; **D** The East China Sea; **E** The Sulu, Celebes, Banda, and Java Seas. **F** The Great Barrier Reef. Scale bars in units of degrees. Observations are also included in the analysis from the East African coast and the Caribbean. Red circles indicate the locations of spawning observations used in the analysis and the extent to which ALAN, SST, and $Kd_{490}$ were averaged for each location. Scaling is different between panels, as indicated by coordinates given in white. Coordinates of spawning locations are provided in a Source Data file.

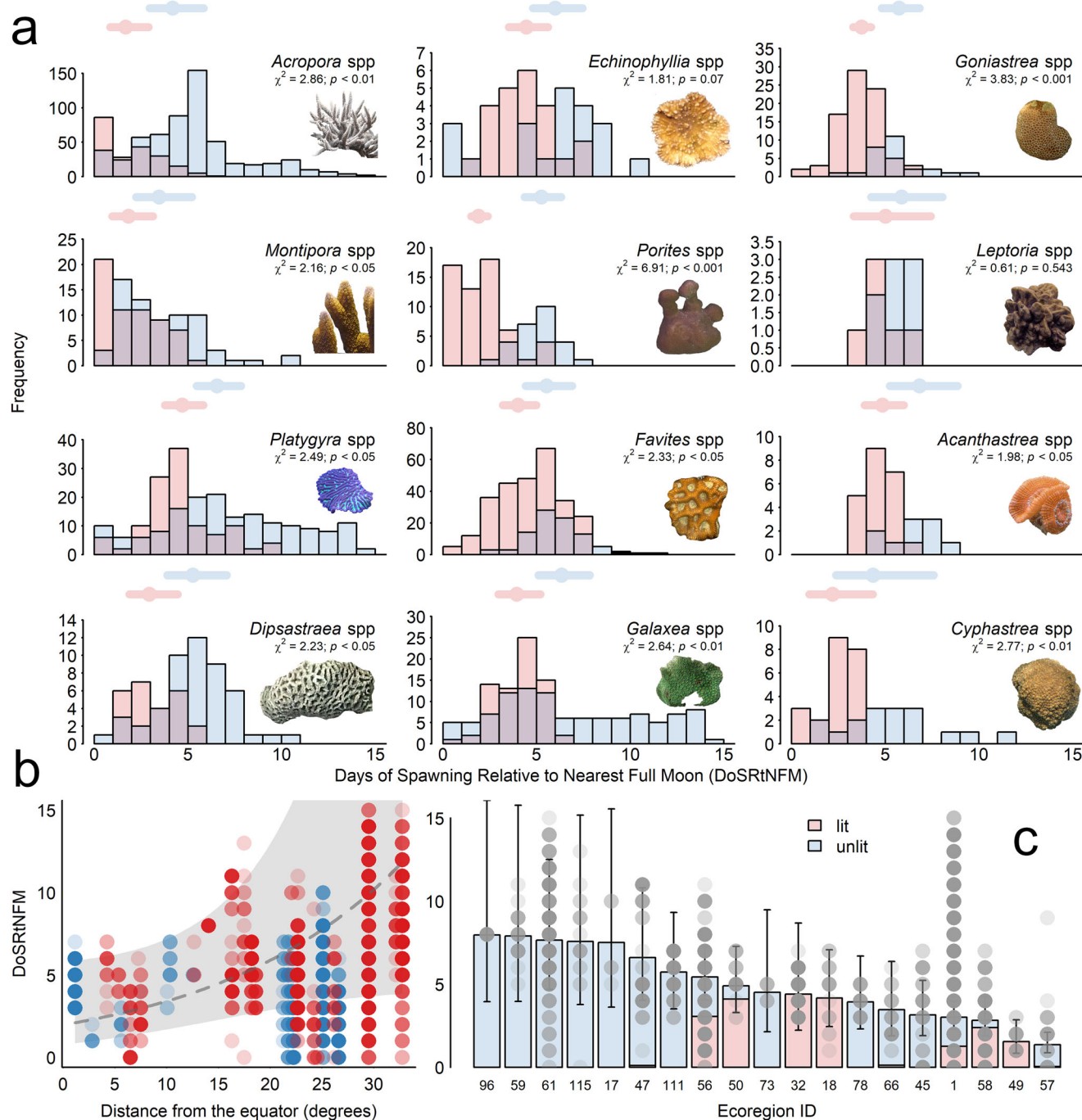

**Fig. 2 | Underwater ALAN disrupts the timing of broadcast spawning by scleractinian corals. a** Frequency histograms of the days spawning is observed relative to full moon (zero on the x-axis) in corals exposed to underwater light pollution (pink) and not exposed (blue). Superimposed points are modeled mean responses ± 95% confidence intervals (Supplementary Table 3). Results are summarized in Supplementary Table 2. The distribution of observations across species within each genus is given in Supplementary Table 4. **b** Corals found at higher latitudes also spawn closer to the full moon. The gray region represents the 95% confidence intervals of the relationship (dashed line). **c** Corals spawned at different times in relation to the full moon in different Ecoregions. Bars are modeled means ± 95% confidence intervals. Colors indicate the number of lit and unlit observations within each ecoregion as a proportion of bar height. Pairwise contrasts between Ecoregions are given in Supplementary Table 5. Ecoregion names are given in Supplementary Table 6. Random effects modeled due to spatial autocorrelation were removed from the prediction of means and confidence intervals in all panels. Raw data is provided as a Source Data file.

(29.5°N) and a low (−5.06°S) latitude coral reef from the broadcast spawning database that were exposed to ALAN (Fig. 3). Changes in irradiance due to solar altitude and lunar phase and altitude were modeled across important months for broadcast spawning (March, April, May, September, October, and November) in the year 2020 (Methods). The ALAN regime was then superimposed to identify the first days following the full moon, where ALAN cuts off the decline in

solar irradiance before the moonrise, creating an extended period of minimum light intensity that provides the trigger for spawning (Fig. 3).

ALAN advances the first day on which an extended duration of minimum light intensity is detectable by one to three days during March, April, May, and November in both the mid and low-latitude reefs, and by four to five days during September and October on the mid-latitude reef (Fig. 3). The timeframe of this advance in the signal

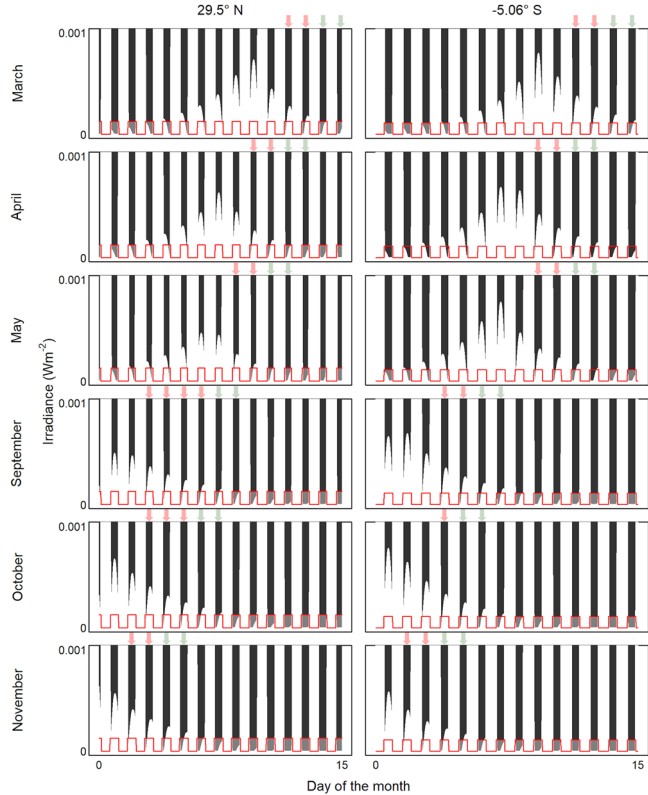

**Fig. 3 | ALAN advances the period of minimum light intensity between sunset and the rise of the full moon that provides the trigger for broadcast spawning.** Natural regimes of darkness (dark gray) at the sea surface are modeled, accounting for solar altitude, and lunar phase and altitude, for a mid-latitude (Eilat Coral Nature Reserve) and a low-latitude (Pulau Bara Lompo) coral reef (Methods). The ALAN regime (red) is assumed to trigger lights on at dusk and off at dawn, with the maximum sea surface irradiance derived from ref. 17. Periods of natural darkness that are masked by ALAN are depicted in light gray. Green arrows indicate the first two nights following the full moon, where a period of natural minimum irradiance (y-axis = 0) between sunset and moonrise is detectable to corals. Red arrows indicate the first nights following the full moon, where ALAN advances the period of minimum light intensity between sunset and moonrise. Raw data is provided as a Source Data file.

for spawning corresponds well with the duration of advancement in spawning observed in the ten impacted coral genera, between 1 and 3 days (Fig. 2).

## Discussion

Contrary to the findings of ref. 10, which suggests reduced precision in coral spawning in the Gulf of Eilat is not related to light pollution but rather temperature elevation alone, we provide evidence that widespread light pollution and not elevated sea surface temperatures advance broadcast spawning likely by affecting the biorhythmicity of physiological processes in corals[21]. While these results present strong evidence that ALAN disrupts coral broadcast spawning at broad spatial scales, they remain derived from observational data. Further manipulative studies are required to establish cause-and-effect relationships between exposure to ALAN and changes in the timing of broadcast spawning in relation to lunar light cycles.

It is widely accepted that the evolutionary benefit of synchronized mass spawning in corals is maximized reproductive contact between gametes[11]. Reproductive contact is maximized firstly by the precision of the spawning event, which results in high concentrations of gametes in the water column. The fertilization success of *Platygura sinensis* gametes declines steadily to almost 0% within 4 h of spawning events due to gamete dilution and aging[22], and spawning one day earlier than

the majority of colonies can reduce fertilization from 49 to 1% in *Montipora digitata*[22]. Secondly, fertilization success and post-fertilization survival are optimized by timing the event during specific windows in the lunar cycle when environmental (e.g. tidal) conditions are optimal. Significant predation of eggs and zygotes by planktivorous (e.g., fish) occurs during coral spawning events[23,24]. Reproductive synchrony is a strategy thought to assist in predator avoidance in many organisms (Predator satiation hypothesis), including corals[25].

While yet to be empirically quantified, advancing the timing of spawning by even one night on artificially lit reefs has potentially profound consequences for coral reproduction. The exposure of coral reefs to biologically important ALAN is variable at fine spatial scales (see Supplementary Fig. 1) such that corals on neighboring lit and unlit reefs may spawn on different nights. This will likely reduce gamete concentrations in the water column for both lit and unlit reefs, potentially reducing fertilization success. Secondly, the temporal partitioning of spawning may lead to reduced fertilization and genetic exchange between lit and unlit reef systems. Finally, lit reefs may be spawning during suboptimal environmental conditions on nights closer to the full moon when tidal currents are stronger and more likely to carry gametes adrift. Combined, these impacts may reduce the reproductive success of corals.

The future of coral reefs remains bleak. Resilient coral reef ecosystems require successful reproduction, dispersal, and recruitment to recover from more frequent and severe mass bleaching and mortality events[26]. Successful reproduction maintains populations and supports evolutionary adaptation to environmental change (e.g., genetic recombination), which enhances fitness[16]. Our results demonstrate that light pollution advances the timing of broadcast spawning, potentially leading to a desynchronization between lit and unlit reef systems in key coral genera. A feasible and effective mitigation measure could be to switch on nighttime lighting in adjacent coastal regions one hour after sunset to ensure the natural dark period between sunset and moonrise that triggers spawning remains detectable, however, the implications of this strategy for safety and the economy may be prohibitive.

## Methods

### Data acquisition

We combined the recently published "Global Atlas of artificial light at night under the sea"[17] (accessed from https://doi.org/10.1594/PANGAEA.929749 on 15/11/2021) with the Coral Spawning Database[18,19] (accessed from https://doi.org/10.25405/data.ncl.13082333.v1 on 25/11/2021), to establish whether broadcast spawning by scleractinian corals is disrupted in artificially lit waters.

At the time it was accessed, the coral spawning database contained 6178 observations recorded at 101 locations around the world between 1978 and 2019, across 330 species representing 61 genera. Observations are recorded both as the calendar day, and the number of days relative to the nearest full moon (DoSRtNF). The 1 km resolution global atlas of artificial light under the sea represents the critical depth to which ALAN can cause biological impacts accounting for (i) the relationship between the new world atlas of artificial night sky brightness[27] and sea surface irradiance[28]; (ii) the wavelength-dependent attenuation of ALAN by the optical properties of the water column[17]; and (iii) a threshold for the sensitivity of marine organisms to light set as those intensities that elicit biological responses in light-sensitive species, *Calanus* copepods[17,28]. The depth of biologically important ALAN represents a single value of light exposure for each pixel, where irradiance would otherwise need to be quoted for multiple depths. The global underwater atlas is available for representative months of the year 2020 such that variability in ALAN penetration due to seasonally variable ocean climatologies is captured, however, interannual trends in ALAN exposure are not. For the

purposes of this analysis, the monthly atlas values were averaged using the raster calculator in QGIS v3.4.8 to provide a representative measure of ALAN exposure across the year 2020, and it was assumed that this atlas was broadly representative of ALAN exposure on coral reefs throughout the 21st century. Mean ALAN exposures across 0.05-degree circular buffers were then calculated and extracted for the coordinates of each spawning observation in the coral spawning database. 4 km resolution global maps of annual average nighttime sea surface temperature and the diffuse attenuation coefficient for downwelling irradiance ($Kd_{490}$, a measure of water clarity) collected by the MODIS-AQUA sensor for the years 2003 to 2022 were downloaded from NASA's Giovanni data portal (https://giovanni.gsfc.nasa.gov/giovanni/) in geoTIFF or the OceanColorWeb level 3 browser in netCDF and imported into QGIS. For each year, average sea surface temperature and $Kd_{490}$ values were extracted across 0.05-degree buffered points representing each spawning sampling site. The rate changes in sea surface temperature and $Kd_{490}$ per year were then quantified for each spawning site using regression to the median to reduce the leverage of outlying data points (CRAN: quantreg).

The analysis was restricted to spawning observations recorded in the 21st century such that the global atlas of artificial light at night under the sea is broadly representative of probable light exposure during the time of observations. This timeframe matches the observed period of broadcast spawning desynchronization in the Gulf of Eliat[10]. Each observation in the coral spawning database was classified as exposed to underwater ALAN ('Lit') where the depth of biologically important ALAN exceeded the depth of the observation or—where no depth was recorded—the minimum depth at which the species is found (from the Coral Trait Database[20], https://coraltraits.org/). Where this was not the case, observations were classified as not exposed to ALAN ('Unlit'). The analysis was performed at the genus level to maximize replication. Genera with less than six recorded observations per ALAN exposure level were excluded from the analysis. Ecoregion was included as a candidate explanatory variable in the analysis to control for potential regional differences in the timing of coral spawning. Data from ecoregions with fewer than six recorded observations were excluded from the analysis. Ex situ observations of coral spawning were omitted from the analysis to remove the influence of nearby light sources from buildings and laboratory light sources. The resulting dataset consisted of 2135 spawning observations recorded at 52 locations in 19 Ecoregions around the world between 2000 and 2019, across 156 species representing 12 genera.

## Analysis

All statistical analyses were performed in R v3.6.1. A spatially autocorrelated generalized linear mixed effects model (CRAN: spaMM) was fitted to the response variable DoSRtNF (positive transformed), accounting for: the effects ALAN exposure ('Lit', 'Unlit'); genus; the rate changes in average annual sea surface temperature ($\Delta$SST, °C) and water clarity ($\Delta Kd_{490}$); the distance of each observation from the equator (DfE, degrees); the interactions of these predictors with ALAN; and the Ecoregion in which observations were made [DoSRtNF~ALAN*Genus + ALAN*$\Delta$SST + ALAN*$\Delta Kd_{490}$ + ALAN*DfE + Ecoregion + Matern(1|longitude+latitude)]. A Poisson error distribution was fitted to account for the right-skewed and dispersed response variable (dispersion = 0.82). The random effects spatial autocorrelation matrix was specified using the Matérn structure of pairwise correlations between the coordinates of each spawning location.

Model selection was performed on all nested versions of the full model. Models were ranked and the most parsimonious model was selected by their value of Akikes Information Criterion (AIC). The significance of the selected model was then validated against a null intercept-only model, and its constituent explanatory terms were interrogated using likelihood ratio tests in a reverse stepwise fashion.

Post hoc interrogation of interactions is not currently supported by CRAN: emmeans for spatially autocorrelated mixed effects models fitted using CRAN:spaMM. Post hoc interrogation of the ALAN*Genus interaction were conducted by fitting independent poisson or negative binomial (when overdispersed) models for each Genus, and performing Turkey's pairwise comparisons using CRAN: multcomp (Supplementary Table 2). The predicted model means and 95% confidence intervals used for plotting in Fig. 2a are given in Supplementary Table 3. P values were not adjusted to avoid the high volume of tests inflating the type 2 error rate. Random effects modeled due to spatial autocorrelation were removed from the prediction of means and confidence intervals. Post hoc pairwise comparisons of the first-order effects of Ecoregion were modeled in the same way.

## Light regime modeling

A generalized modeling framework available from https://github.com/timjsmyth/TidalLight[29] was developed using the Python programming language in order to simulate the surface spectral light field as a function of location (latitude, longitude), date, and time[17]. The model was run at a 5-min timestep resolution for four locations for the date periods 1 March–31 May 2020 and 1 September–30 November 2020.

## Solar spectral model

The top of atmosphere (TOA) spectral solar irradiances, $E_0(\lambda)$, at 1 nm resolution were extracted from a lookup-table[30] of the solar spectral irradiance, $H_0(\lambda)$, and corrected for the eccentricity ($\varepsilon$) of Earth's orbit (function of day of year, D) using the equation:

$$E_0(\lambda) = H_0(\lambda)\left(1 + \varepsilon\cos\left\{\frac{2\pi(D-3)}{365}\right\}\right)^2 \quad (1)$$

where $\varepsilon$ is 0.0167. The Gregg and Carder[31] spectral marine atmosphere model was used to determine the spectral (just) above surface solar irradiance, assuming clear sky conditions. This is a relatively simple model but does take into consideration gaseous absorption and aerosol optical properties and allows for the partitioning of the irradiance field into direct, $E_{dd}(\lambda)$, and diffuse, $E_{ds}(\lambda)$, components. The atmospheric parameters used in the model are shown in Supplementary Table 7. The global above surface spectral irradiance is the sum of these two terms:

$$E_d(\lambda, 0^+) = E_{dd}(\lambda) + E_{ds}(\lambda) \quad (2)$$

Additionally, the spectral twilight model of ref. 32 was used to determine the spectral sky (diffuse) irradiance for solar zenith angles ($\theta_z$) between 0° and −18°, using a lookup-table constructed from their rural sky observations (Cherry Springs State Park). This allowed the twilight period to be split between civil ($0° \leq \theta_z < -6°$), nautical ($-6° \leq \theta_z < -12°$), and astronomical ($-12° \leq \theta_z < -18°$) partitions. The spectrally resolved rural sky observations are the best available analog to remote coastal marine locations. Additionally, our investigations are limited to clear sky cases, further reducing the potential for distant urban sky glow to bias the rural observations on cloudier nights.

Our model neglects the second-order effects of air glow, zodiacal light, and integrated starlight and considers them as nighttime background levels, uninfluenced by Lunar and ALAN sources. Assuming dark-sky brightness of 22 mag arcsecond$^{-2}$ (ref. 33), this is equivalent to $1.712 \times 10^{-4}$ cd m$^{-2}$, which is around 0.25 μW m$^{-2}$ which is of second-order importance to lunar and ALAN sources.

## Lunar spectral model

The TOA spectral lunar irradiances were determined using the TOA spectral solar irradiances (Eq. 1), spectrally varying lunar albedo (Table 2 of ref. 34), and a lunar semi-diameter view angle of 0.26°. The lunar zenith angle was calculated as a function of location, date, and time

using the Python astropy package. The phase curve of ref. 35 was used to account for the full moon brightening, and the lunar phase was calculated as a function of latitude, time, and date using the Python astroplan package. The Gregg and Carder[31] model was then used to determine the spectral surface lunar irradiance, assuming clear sky conditions (see Supplementary Table 7).

The substantive differences between the solar and lunar components are (a) the magnitude of the TOA irradiance (Eq. 1), which for the lunar component is roughly five orders of magnitude less than the solar for a full moon; (b) the phase of the moon; and (c) their respective celestial geometries (i.e., position in the sky).

### ALAN source term

The above surface ALAN spectral irradiances can be generally derived from spectral shape functions for a given lighting source (e.g., light emitting diode (LED), high-pressure sodium (HPS), low-pressure sodium (LPS)) and scaled by a reasonable/informed factor to give intensity. In this paper, we use the approach outlined in ref. 17 validated against data originally reported in ref. 36 where the above surface sky brightness (ref. 27) for a given location is spectrally resolved into blue (400–500 nm), green (495–560 nm), and red (640–720 nm) broad wavelength bands based on empirical field data collected close to the city of Plymouth, UK[28].

### Reporting summary

Further information on research design is available in the Nature Portfolio Reporting Summary linked to this article.

## Data availability

The global atlas of artificial light at night under the sea[37] is available to download from https://doi.org/10.1594/PANGAEA.929749. The coral spawning database[18] is available from https://doi.org/10.25405/data.ncl.13082333.v1. The data used in the analysis are provided in the Source Data file. Source data are provided with this paper.

## Code availability

The code used to generate the light cycles[38] in Fig. 3 is available at https://doi.org/10.5281/zenodo.7777966.

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

## Acknowledgements

This work was supported by the Natural Environment Research Council (grant numbers NE/S003533/2 and NE/S003568/1 awarded to T.W.D. and T.S.)

## Author contributions

T.W.D. developed the idea and methodological approach; T.S. imple-mented the hydrological optics modeling; T.W.D. extracted and ana-lysed the data, developed the figures, and the first draft of the MS. T.W.D., O.L., S.T., L.F.d.B.M., J.W., C.D.A., and T.S. wrote the second draft of the MS. T.W.D., T.S., J.W., and C.D.A. secured funding that supported the research.

## Competing interests

The authors declare no competing interests.
