## [Peer Review File · Nature Communications]

Global disruption of coral broadcast spawning associated with artificial light at nightEditorial Note: This manuscript has been previously reviewed at another journal that is not operating a transparent peer review scheme. This document only contains reviewer comments and rebuttal letters for versions considered at *Nature Communications*.

Reviewer #2 (Remarks to the Author):

Davies Review 2

This is my second review of this manuscript. The authors have done a really good job of addressing comments from the reviewers and this is a much better paper as a result. I still think they overstate the potential implications of their findings, particularly in relation to coral recruitment and have made some suggestions for improvement.

Abstract

Line 17: "certain nights of the year following the full moon" this statement, as a generalisation is true (i.e., many species spawn a few days after full moons), but many also spawn before the nearest full moon or spawn during the day or spawn 1-2 weeks after full moon (closer to new moon). So I wonder if here (and elsewhere in the manuscript) it would be more accurate to say that spawning happens predictably in relation to lunar cycles for a given location and species. For example, if you look at the CSD it shows that spawning night relative to nearest full moon ranges from 15 days before to 14 days after full moon. The majority (about 50%) of observations do occur in the five days following full moon, but >30% occur from 6-14 days after full moon and around 15% between full moon and 15 days before. Note there is almost certainly a bias in the data in that people tend to look for spawning after full moon, so many spawning events may occur at other times and go unobserved.

Lines 18-21: I think these two sentences set the study up in the wrong way. The way this reads to me is that there is evidence that broadcast spawning is increasingly being disrupted but we don't know exactly what is causing it. In reality, we know nothing about the extent that broadcast spawning has been disrupted because we simply do not have sufficient time series data for most locations. I think it would be much better to say that human disturbances such as light pollution have the potential to disrupt rhythms, but to date this has not been tested on a global scale. You say that in the next sentence which is great, but make it about "threatening" coral reef health rather than just focusing on disrupting rhythms (which may in turn have ecological impacts).

Lines 29-31: In my earlier review of this paper I was critical about the authors making the link between shifts in spawning timing and recruitment and connectivity. I just want to reiterate that as far as I know, there are no coral studies in existence showing a negative impact of changes in spawning timing on recruitment success. The authors need to be careful not to make this implicit link between shifts in spawning timing and other ecological processes without any actual data to support their statements. Recruitment of new corals is governed by a complex series of factors including (ocean currents, properties of the substratum, predation, grazing rates, benthic competition etc.), all of which can influence recruitment success. Typically, coral "recruits" only show up on the benthos when they are 1 or 2 years old, so much can happen in between spawning and recruitment. I think the authors need to rephrase this to say that shifts in reproductive timing has implications for other ecological processes involved in persistence and resilience of reefs, but not to make any assumptions about what these implications are.

Introduction:

Line 40: I would really like the authors to define what they mean by "synchrony" as "desynchronized" these terms are used throughout, but no working definition is provided. Synchrony is not a straightforward concept when it comes to coral reproduction and means different things to different people. It is rarely quantified in studies of coral spawning and indeed the study you cite (refs 10 and 11) did not actually measure or quantify synchrony (which is the proportion of the population reproducing at the same time). I'd really like to see the authors provide an actual definition of synchrony and be clear what they mean when saying that "desynchronisation" has been found.

Line 46: perhaps be clear here and say that these observations are from 1 or 2 sites in the extreme north of the Red Sea, this has not been shown from the Red Sea in general.

Lines 47-48: I'd suggest deleting the last part of this sentence (from leading to the assertion...). As I said in my earlier review, the paper you cite does not show a causal link between spawning disruption and changes in recruitment. Your study is not about recruitment, rather it is about factors disrupting spawning timing, so I see no reason to keep this in the introduction as you don't go on to support or challenge this assertion anywhere in your paper.

Line 51: A potential threat to coral reproduction, maybe, but recruitment...we just don't know, there's no data on this so why make this link when it's not even part of your study. I think if you are going to talk about recruitment then you need to define it clearly (recruitment is not the same as coral settlement which happens soon after spawning). My feeling is that references to recruitment should be dropped throughout the manuscript as you have no data on recruitment and we have no idea whatsoever about the link (if any) between shifts in spawning timing and recruitment success.

Line 130-132: See my comments earlier. It's true that many species do spawn a few days after full moon, but many also spawn on or before full moon and some key species (e.g., *Porites rus*) spawn early in the morning.

Line 137: See my earlier comments. Consequences for reproduction yes, but not necessarily recruitment. This is just pure speculation without considering all of the other processes that affect recruitment.

Line 140: "elevating predator risk" again, this is total speculation. It may actually reduce predation if fish are expecting corals to spawn at other times. In reality, we have absolutely no idea what the link is between egg consumption and reproductive success and the theory of synchrony for predator avoidance had never been tested for corals.

Lines 145-152: I was really confused by this paragraph. How have you shown that elevated SSTs independently advance spawning timing? Also, be clear here on what Shlesinger et al actually showed. They did NOT demonstrate spawning failure, rather they found that for 2 species in Eilat, spawning was more spread out than previously thought. That isn't failure to spawn it's just spawning occurring over more nights than expected.

Lines 159-161: While it's good to try and suggest some management measures, I think you have to be careful about being prescriptive here, particularly considering that we don't really know what the ecological impact of these findings are. Changing lighting timing of lighting potentially has all sorts of implications for humans (e.g., for safety) and would likely be highly context specific. Perhaps a better suggestion would be to promote further research to understand what the ecological impacts are (how serious are they) or to carry out trials in certain locations to do as suggested and switch on lights later (where it is feasible/safe to do so) to see the impact on spawning timing.

Reviewer #3 (Remarks to the Author):

I am happy with the responses to my concerns.

Reviewer #4 (Remarks to the Author):

The authors have adequately addressed my comments in this updated version. In this new version they have used the MODIS AQUA SST Night time series. Personally, I believe the MODIS actually brings the sea surface temperature closer to the coastline in situ situations due to its high resolution. Yet I understand why the authors chose to remove the MERRA-2 SST input after going through their response to Reviewer #2. The paper is overall much improved and I have no more question on the SST inputs.

Reviewer #5 (Remarks to the Author):

The manuscript entitled "Global disruption of coral broadcast spawning by artificial light at night" by Davies et al analysed the potential effects of artificial light at night (ALAN) on the timing of spawning of corals across the globe.

I think the paper is extremely interesting and the subject definitely warrants investigation, however, I think there are some issues with how the analyses were done. My main concern is that data from time of spawning covers almost two decades while there is only one year of data on light. Also, 'Time' doesn't seem to be a factor in the models, so how did you consider that? Similarly, I'm not sure how ecologically/biologically relevant is the fact that temperature (and e.g. turbidity data) has been averaged across years. One would expect that, if temperature is related to changes in timing of spawning, then this would have potentially worsened with time (i.e. as temperature rises), however the way the model has been done, it is impossible to assess it or consider this in relation to effects of ALAN, potentially affecting the interpretation of results. In any case, I do commend the authors for the amount of work they have done. I think authors addressed comments from reviewer 1 appropriately and the text is now much clearer. More details are still needed in how the spatial autocorrelation was considered in the models. For example, which random factors were included? It would be good as well to know how many species, on average, are considered within the factor 'genera'. For example does one genus have 50 species being considered in the analyses while another genus have only a couple? Also, not being a coral expert myself, it would be good to have a few sentences discussing whether species within the same genera spawn at the same time...

Minor comments:

Lines 124-125 – Need references

Lines 137 – This sentence is confusing. Please re-write for clarity.

Lines 139 – Have you seen that? You have the data, you could potentially check to see whether that has happened? At least in a few cases...

Reviewer #2 (Remarks to the Author):

- This is my second review of this manuscript. The authors have done a really good job of addressing comments from the reviewers and this is a much better paper as a result. I still think they overstate the potential implications of their findings, particularly in relation to coral recruitment and have made some suggestions for improvement.

RESPONSE: We are very pleased that reviewer two finds that we have done a good job of addressing the reviewer comments and that the manuscript is much improved. We would like to thank them again for their detailed and insightful comments on the first submission. We did make considerable edits to the first submission in order to address the reviewers' concerns around overstating the potential implications of our findings. We have now made additional edits to address the concerns they outline below.

- Abstract
Line 17: "certain nights of the year following the full moon" this statement, as a generalisation is true (i.e., many species spawn a few days after full moons), but many also spawn before the nearest full moon or spawn during the day or spawn 1-2 weeks after full moon (closer to new moon). So I wonder if here (and elsewhere in the manuscript) it would be more accurate to say that spawning happens predictably in relation to lunar cycles for a given location and species. For example, if you look at the CSD it shows that spawning night relative to nearest full moon ranges from 15 days before to 14 days after full moon. The majority (about 50%) of observations do occur in the five days following full moon, but >30% occur from 6-14 days after full moon and around 15% between full moon and 15 days before. Note there is almost certainly a bias in the data in that people tend to look for spawning after full moon, so many spawning events may occur at other times and go unobserved.

RESPONSE: We agree. From our understanding of the dataset, broadcast spawning can for whatever reason be more variable in timing than our statement implies. We have therefore edited the statement here and elsewhere as reviewer 2 suggests.

- Lines 18-21: I think these two sentences set the study up in the wrong way. The way this reads to me is that there is evidence that broadcast spawning is increasingly being disrupted but we don't know exactly what is causing it. In reality, we know nothing about the extent that broadcast spawning has been disrupted because we simply do not have sufficient time series data for most locations. I think it would be much better to say that human disturbances such as light pollution have the potential to disrupt rhythms, but to date this has not been tested on a global scale. You say that in the next sentence which is great, but make it about "threatening" coral reef health rather than just focusing on disrupting rhythms (which may in turn have ecological impacts).

RESPONSE: We have removed these sentences from the abstract, so that it is focused on ALAN rather than the disruption itself.

- Lines 29-31: In my earlier review of this paper I was critical about the authors making the link between shifts in spawning timing and recruitment and connectivity. I just want to reiterate that as far as I know, there are no coral studies in existence showing a negative impact of changes in spawning timing on recruitment success. The authors need to be careful not to make this

implicit link between shifts in spawning timing and other ecological processes without any actual data to support their statements. Recruitment of new corals is governed by a complex series of factors including (ocean currents, properties of the substratum, predation, grazing rates, benthic competition etc.), all of which can influence recruitment success. Typically, coral “recruits” only show up on the benthos when they are 1 or 2 years old, so much can happen in between spawning and recruitment. I think the authors need to rephrase this to say that shifts in reproductive timing has implications for other ecological processes involved in persistence and resilience of reefs, but not to make any assumptions about what these implications are.

RESPONSE: We understand the reviewers concerns. We did make edits to the first submission which we hoped would adress these, particularly by introducing clauses such as ‘could’ in this statement to clearly indicate that it was a statement based on inference rather than evidence. Nonetheless we accept the reviewers concerns. We do now cite empirical evidence in the Discussion to support the first part of this statement: “*Advancing the timing of mass spawning could decrease the probability of gamete fertilization and survival*”. We have then edited the second part of this statement in line with the reviewers suggestion.

- Introduction:
- Line 40: I would really like the authors to define what they mean by “synchrony” as “desynchronized” these terms are used throughout, but no working definition is provided. Synchrony is not a straightforward concept when it comes to coral reproduction and means different things to different people. It is rarely quantified in studies of coral spawning and indeed the study you cite (refs 10 and 11) did not actually measure or quantify synchrony (which is the proportion of the population reproducing at the same time). I’d really like to see the authors provide an actual definition of synchrony and be clear what they mean when saying that “desynchronisation” has been found.

RESPONSE: We have now edited the statement to be inclusive of the reviewers suggested definition. “*Synchronized broadcast spawning is a reproductive strategy that involves the release of a large number of eggs and sperm simultaneously by a large proportion of a local coral population on specific nights of the year.*”

- Line 46: perhaps be clear here and say that these observations are from 1 or 2 sites in the extreme north of the Red Sea, this has not been shown from the Red Sea in general.

RESPONSE: We are now specific that the observations were made in the northern red sea.

- Lines 47-48: I’d suggest deleting the last part of this sentence (from leading to the assertion...). As I said in my earlier review, the paper you cite does not show a causal link between spawning disruption and changes in recruitment. Your study is not about recruitment, rather it is about factors disrupting spawning timing, so I see no reason to keep this in the introduction as you don’t go on to support or challenge this assertion anywhere in your paper.

RESPONSE: We have removed the later half of this statement as suggested.

- Line 51: A potential threat to coral reproduction, maybe, but recruitment...we just don’t know, there’s no data on this so why make this link when it’s not

even part of your study. I think if you are going to talk about recruitment then you need to define it clearly (recruitment is not the same as coral settlement which happens soon after spawning). My feeling is that references to recruitment should be dropped throughout the manuscript as you have no data on recruitment and we have no idea whatsoever about the link (if any) between shifts in spawning timing and recruitment success.

RESPONSE: We have removed reference to recruitment here.

- Line 130-132: See my comments earlier. It's true that many species do spawn a few days after full moon, but many also spawn on or before full moon and some key species (e.g., *Porites rus*) spawn early in the morning.

RESPONSE: We have now edited this statement to read: "*Secondly, fertilization success and post fertilization survival are optimised by timing the event during specific windows in the lunar cycle when environmental (e.g. tidal) conditions are optimal.*"

- Line 137: See my earlier comments. Consequences for reproduction yes, but not necessarily recruitment. This is just pure speculation without considering all of the other processes that affect recruitment.

RESPONSE: We have removed reference to recruitment here.

- Line 140: "elevating predator risk" again, this is total speculation. It may actually reduce predation if fish are expecting corals to spawn at other times. In reality, we have absolutely no idea what the link is between egg consumption and reproductive success and the theory of synchrony for predator avoidance had never been tested for corals.

RESPONSE: We have removed reference to predation risk.

- Lines 145-152: I was really confused by this paragraph. How have you shown that elevated SSTs independently advance spawning timing? Also, be clear here on what Shlesinger et al actually showed. They did NOT demonstrate spawning failure, rather they found that for 2 species in Eilat, spawning was more spread out than previously thought. That isn't failure to spawn it's just spawning occurring over more nights than expected.

RESPONSE: Thanks for pointing this out. Note that in the reanalysis we have conducted in response to reviewer 5, the annual average SST variable has been replaced with one that captures the rate of change in SST over time. This was not a significant predictor of broadcast spawning in relation to full moon and so reference to SST has been removed from the analysis.

- Lines 159-161: While it's good to try and suggest some management measures, I think you have to be careful about being prescriptive here, particularly considering that we don't really know what the ecological impact of these findings are. Changing lighting timing of lighting potentially has all sorts of implications for humans (e.g., for safety) and would likely be highly context specific. Perhaps a better suggestion would be to promote further research to understand what the ecological impacts are (how serious are they) or to carry out trials in certain locations to do as suggested and switch on lights later (where it is feasible/safe to do so) to see the impact on spawning timing.

RESPONSE: The reviewer makes a good point. We feel it is important to make some recommendation for how impacts can be avoided. However we also recognise

need to balance such suggestions against their implications for society. We have now edited the statement to read “*A feasible and effective mitigation measure could be to switch on nighttime lighting in adjacent coastal regions one hour after sunset to ensure the natural dark period between sunset and moonrise that triggers spawning remains detectable, however the implications of this strategy for safety and the economy may be prohibitive.*”

Reviewer #3 (Remarks to the Author):

- I am happy with the responses to my concerns.

RESPONSE: We are pleased that reviewer three is happy with our responses, and thank them again for their insightful comments.

Reviewer #4 (Remarks to the Author):

- The authors have adequately addressed my comments in this updated version. In this new version they have used the MODIS AQUA SST Night time series.
- Personally, I believe the MODIS actually brings the sea surface temperature closer to the coastline in situ situations due to its high resolution. Yet I understand why the authors chose to remove the MERRA-2 SST input after going through their response to Reviewer #2. The paper is overall much improved and I have no more question on the SST inputs.

RESPONSE: We are pleased that reviewer 4 finds the manuscript is much improved and has raised no further comments. The introduction of the MODIS-AQUA data into the analysis has now enabled us to look at the effect of the rate of change in SST and Kd490 in response to reviewer 5, so we are glad that reviewer 4's comments prompted us to make this move.

Reviewer #5 (Remarks to the Author):

- The manuscript entitled “Global disruption of coral broadcast spawning by artificial light at night” by Davies et al analysed the potential effects of artificial light at night (ALAN) on the timing of spawning of corals across the globe. I think the paper is extremely interesting and the subject definitely warrants investigation, however, I think there are some issues with how the analyses were done. My main concern is that data from time of spawning covers almost two decades while there is only one year of data on light. Also, ‘Time’ doesn’t seem to be a factor in the models, so how did you consider that? Similarly, I’m not sure how ecologically/biologically relevant is the fact that temperature (and e.g. turbidity data) has been averaged across years. One would expect that, if temperature is related to changes in timing of spawning, then this would have potentially worsened with time (i.e. as temperature rises), however the way the model has been done, it is impossible to assess it or consider this in relation to effects of ALAN, potentially affecting the interpretation of results.

RESPONSE: We are very grateful to Reviewer 5 for their comments which have prompted a reanalysis of the data. Before outlining the changes we have made, we would first like to address the point about time directly.

The global atlas of artificial light at night under the sea is derived from the new world atlas of artificial sky brightness published by Falchi et al in 2016. That atlas takes satellite imagery of the earth's night time lights from the VIIRS sensor and applies an atmospheric dispersion model to account for the skyglow effect of artificial light. It represents the extent of ALAN in a fixed time point and as such we are unable to measure or model ALAN under the sea at global scale for multiple points in time. Our analysis is then a spatial analysis. The analysis was restricted to spawning observations recorded in the 21st century such that the global atlas of artificial light at night under the sea is broadly representative of probable light exposure during the time of spawning observations.

We do accept however that while the analysis is primarily spatial, some of our input variables could better capture temporal components (for example changes in SST over time as reviewer 5 suggests). We have then reanalysed the data using the rate changes in STT and Kd_{490} over time rather than the mean of the variables over the whole time period. The rate changes were derived for each spawning location as the coefficient of the relationship between annual average SST (or Kd_{490}) and year using regression to the median to reduce the influence of unusually warm or cold years. Interestingly, the rate change variables have no quantifiable effect on the timing of broadcast spawning, at least within the lunar month.

- In any case, I do commend the authors for the amount of work they have done. I think authors addressed comments from reviewer 1 appropriately and the text is now much clearer. More details are still needed in how the spatial autocorrelation was considered in the models. For example, which random factors were included? It would be good as well to know how many species, on average, are considered within the factor 'genera'. For example does one genus have 50 species being considered in the analyses while another genus have only a couple? Also, not being a coral expert myself, it would be good to have a few sentences discussing whether species within the same genera spawn at the same time.

RESPONSE: We are very pleased that reviewer 5 has commended us for the work that we have put into the first revision.

Spatial autorrelation specification

We have now also added further detail about the random effects specification for the spatial autorellation in the model as follows:-

"The random effects spatial autocorrelation matrix was specified using the Matérn structure of pairwise correlations between the coordinates of each spawning location."

We have also added the random effects terms to models quoted in the results and methods, for example [$DoSRtNF \sim ALAN * Genus + ALAN * \Delta SST + ALAN * \Delta Kd_{490} + ALAN * DfE + Ecoregion + Matern(1 | longitude + latitude)$].

We have also added a statement to the legend of extended data table 1 which reads *"All models were fitted with a random effects term to account for spatial autocorrelation specified using the Matérn structure of pairwise correlations between the coordinates of each spawning location [+ Matern(1 | longitude + latitude)]."*

Species representation across genera.

We have included a new table to the extended data (Extended data table 4), which provides the breakdown of observations across species within each genera.

- Minor comments:]
- Lines 124-125 – Need references

RESPONSE: We have now added a citation for the statement.

- Lines 137 – This sentence is confusing. Please re-write for clarity.

RESPONSE: We have reformulated the sentence starting on line 137 of the initial submission.

- Lines 139 – Have you seen that? You have the data, you could potentially check to see whether that has happened? At least in a few cases.

RESPONSE: Unfortunately the coral spawning database site locations are not sufficient in number or spatial coverage to present many opportunities for direct comparisons between neighbouring lit and unlit sites. We have looked at several potential candidate locations where there does appear to be neighbouring sampling locations (Sesoko, Qita al Kirsh, Chumphon and Magnetic Island). Unfortunately in each location either all of the observations are from corals that the analysis defined as lit, or the data coverage is insufficient to facilitate comparison.

Reviewer #5 (Remarks to the Author):

The authors have addressed my initial comments re the analyses and the inclusion of temporal aspect of variables such as temperature, and SST. I'm still concern about some overstatements throughout the manuscript and would like to see some acknowledgement that results are mainly relationships (based on patterns) between timing of coral spawning and ALAN, and manipulative studies need to be done to establish cause-effect relationships. Furthermore, it would be good to clearly stated limitations of the study, including the lack of temporal data on ALAN, which restricts the possible analyses. Other than that, I appreciate the work done by the authors and have no further comments.

Reviewer #5 (Remarks to the Author):

The authors have addressed my initial comments re the analyses and the inclusion of temporal aspect of variables such as temperature, and SST. I'm still concern about some overstatements throughout the manuscript and would like to see some acknowledgement that results are mainly relationships (based on patterns) between timing of coral spawning and ALAN, and manipulative studies need to be done to establish cause-effect relationships. Furthermore, it would be good to clearly stated limitations of the study, including the lack of temporal data on ALAN, which restricts the possible analyses. Other than that, I appreciate the work done by the authors and have no further comments.

RESPONSE: The authors are very pleased that reviewer 5 is satisfied that our reanalysis has addressed their concerns. Reviewer 5 has requested that we are clear about the limitations of the study, including recommending manipulative experiments to establish cause and effect and being more explicit about the ALAN variable being representative of one year (2020).

We have added a statement to the end of the first paragraph of the discussion that we hope satisfies their first suggestion:-

“While these results present strong evidence that ALAN disrupts coral broadcast spawning at broad spatial scales, they remain derived from observational data. Further manipulative studies are required to establish cause and effect relationships between exposure to ALAN and changes in the timing of broadcast spawning in relation to lunar light cycles.”

We have also made additional edits to the methods making it more explicit that the ALAN variable is derived from an atlas of ALAN exposure representative of the year 2020.

“The global underwater atlas is available for representative months of the year 2020 such that variability in ALAN penetration due to seasonally variable ocean climatologies is captured, however interannual trends in ALAN exposure are not. For the purposes of this analysis the monthly atlas values were averaged using the raster calculator in QGIS v3.4.8 to provide a representative measure of ALAN exposure across the year 2020, and it was assumed that this atlas was broadly representative of ALAN exposure on coral reefs throughout the 21st century.”